# MASE: An Instrument Designed to Record Underwater Soundscape

**DOI:** 10.3390/s22093404

**Published:** 2022-04-29

**Authors:** Iván Rodríguez-Méndez, Jonas Philipp Lüke, Fernando Luis Rosa González

**Affiliations:** Department of Industrial Engineering, University of La Laguna, P.O. Box 456, 38200 La Laguna, Spain; jpluke@ull.edu.es (J.P.L.); frosa@ull.edu.es (F.L.R.G.)

**Keywords:** passive acoustic monitoring, PAM, hydrophone, underwater noise monitoring, scientific buoy, remote processing, Raspberry Pi, underwater technology, subsea monitoring, ecoacoustic indices

## Abstract

The study of sound in the natural environment provides interesting information for researchers and policy makers driving conservation policies in our society. The soundscape characterises the biophony, anthrophony and geophony of a particular area. The characterisation of these different sources can lead to changes in ecosystems and we need to identify these parameters in order to make the right decision in relation to the natural environment. These values could be extrapolated and potentially help different areas of ecoacoustic research. Technological advances have enabled the passive acoustic monitoring (PAM) of animal populations in their natural environment. Recordings can be made with little interference, avoiding anthropogenic effects, making it a very effective method for some species such as cetaceans and other marine species in addition to underwater noise studies. Passive acoustic monitoring can be used for population census, but also to understand the effect of human activities on animals. However, recording data over long periods of time requires large storage and processing capacity to handle all the acoustic events generated. In the case of marine environments, the installation of sensors and instruments can be costly in terms of money and maintenance effort. In addition, if they are placed offshore, a data communication problem arises with coverage and bandwidth. In this paper, we propose a low-cost instrument to monitor the soundscape of a marine area using ecoacoustic indices. The instrument is called MASE and provides three echo-acoustic indices at 10 min intervals that are available in real time, which drastically reduces the volume of data generated. It has been operating uninterruptedly for a year and a half since its deployment, except during maintenance periods. MASE has been able to operate uninterruptedly, and maintain an adequate temperature inside while preserving its structural integrity for long periods of time. This has allowed the monitoring and characterisation of the soundscape of the test area in Gando Bay, Gran Canaria Island (Spain) without the need for human intervention to access the data on the instrument itself. Thanks to its integration with an external server, this allows the long-term monitoring of the soundscape, and it is possible to observe changes in the soundscape. In addition, the instrument has made it possible to compare the period of acoustic inactivity during confinement and the return of anthropogenic acoustic activity at sea.

## 1. Introduction

The study of sound in the natural environment provides interesting information for researchers and politicians who push conservation policies in our society. The soundscape characterises the biophony, anthrophony, and geophony of a specific area. The characterisation of these different sources can cause changes in ecosystems and we need to identify those parameters in order to make the right decision in relation with the natural environment. These values could be extrapolated and potentially help different areas of ecoacoustic research.

Over the years, the introduction of activities that generate anthrophony in the ocean has increased, leading to changes in ecosystems. Anthropogenic noise can cause significant changes in marine habitats and the organisms present in them by changing their behaviour [1,2,3,4,5]. In fact, Directive 56/2008 of the European Marine Strategy list several different protected areas and respective harmful energy levels giving the qualitative descriptors to determine a good environmental status [6]. The Marine Strategy Framework Directive requires that European Member States implement strategies toward a good environmental status in European seas. From the perspective of monitoring underwater ambient noise, it is necessary to assess the 11.2.1 indicator, which is described through the average values of sound pressure (re 1 μPa RMS) over a year, in the two one-third octave bands centred on 63 and 125 Hz [7]. The sources of noise are commonly related to transport, mining, fishing and construction. In addition, recreation, industries and tourism can affect coastal waters.

In existing regulations, noise sources are mainly categorised into two types: high-intensity impulse noise and low-frequency continuous noise [6,8]. High-intensity impulse noise is generated in activities such as pile driving or active sonar use. Noise related to pile driving is generated in constructions under the surface because it requires large machines that reach sound levels of up to 260 dB per 1 μPa, without noise reduction actions [8,9]. Commercial, fishing and research vessels generate the noise associated with active sonars used for object detection [8,10]. Sources of continuous low-frequency noise are associated with ships and large vessels in addition to recreation boats and whale-watching ships. Maritime traffic generally does not produce intense noise; however, acoustic pollution is introduced over time, which can damage entire animal populations. The noise generated inside the ships is caused by mechanical vibrations from engines, generators, or propeller rotation. The frequency range for this type of sound is between 0.1 and 1 kHz [4].

Although studies have been conducted on the introduction of noise into the ocean, there is little understanding of the noise thresholds that are hazardous to the marine biome in terms of frequency and energy.

In recent years, technological advances have enabled the passive acoustic monitoring (PAM) of animal populations in their natural environment. Recordings can be made with little interference, avoiding anthropogenic effects, which makes it a very effective method for some species such as cetaceans. Passive acoustic monitoring can be used to conduct a population census as well as to understand the effect of human activities on the animals. However, recording data over long periods of time requires extensive storage and processing capacity to manage all generated acoustic events. In most cases, the manual processing of acoustic data is a time-consuming task. This requires the automation of this task. In the case of marine environments, the installation of sensors and instruments can be expensive in terms of money and maintenance efforts. Furthermore, if they are placed offshore, a data communication problem arises with coverage and bandwidth. A strategy to tackle this is to apply data reduction techniques.

Ecoacoustic indices characterise soundscapes with a reduced set of parameters that allow the discrimination of different sound sources in the environment covered [11,12,13,14,15,16,17] and therefore can be used to optimise data generation. These enable the evaluation of soundscape degradation due to human activities and the detection of changes in the soundscape, such that they could be a useful tool to make decisions about conservation in form of macro-indicators. Since ecoacoustic indices were first introduced to the community by [11,12], many studies have been carried out on the development and application of such indices. Many of these studies have been conducted on the use of acoustic indices on avian communities, as well as some works that exist about their use in marine environments [18]. Indices summarise the content of an audio signal over a certain interval of time. In most studies, this time is established at one minute, five minutes, or ten minutes. Furthermore, indices can cover certain frequency ranges or even the whole spectrum. The computation of acoustic indices over time may provide interesting information about changes in the soundscape of the recording area as they show changes in the richness of species in that area [15,19]. The authors of [19] distinguished two groups of indices: α-indices and β-indices.

The first ecoacoustic studies made recordings at certain locations in a study area for a period of time [20]. Later, these recordings must be recovered and processed off-line by using an ecoacoustic software [21,22,23,24]. The recordings generated by such instruments can also be used for underwater noise characterisation and modelling [25,26]. To cover longer periods of time, the approach has mostly not been recording the whole audio stream. Instead, a short recording is made with a certain periodicity in time. For example, in [12], the authors recorded one minute every 11 min, while in [27], the authors recorded ten minutes at the beginning of each hour and every half hour. Such limitations were introduced by the need to save storage when the study is carried out over a long period of time. There are some commercial products on the market with the necessary features [28,29]. To overcome the limitation of storage capacity, the recording system could be provided with a processing hardware that enables processing the raw audio signal and only store ecoacoustic indices. The Soundscape Explorer instrument provided by Lunilettronik [30], which computes ACI [13] in real time and is used in [17], is an example of this approach. In [31], a smart low-cost acoustic device for monitoring biodiversity is proposed. In many cases, the limitation of delayed data recovery remains. However, this could be overcome by adding a communication system to the instrumentation that enables recovering data while they are captured. Examples of such an approach can be found in [32,33]. Another point that must be considered is the power consumption of the instrumentation because many devices are powered by batteries that limit the operation time, although in some cases, they are also recharged with solar panels. Most of these developments are for land environments. Obviously, the transportation to the underwater environment is more difficult. The work presented in [34] showed such an adaptation, with design ideas very similar to those presented in this paper.

In this paper, we propose a low-cost instrument for monitoring the soundscape in a marine area using ecoacoustic indices. The instrument is called MASE and provides three ecoacoustic indices that will be described in Section 2.3 at intervals of 10 min that are available in real time. The main objective of MASE is to study the presence of anthropogenic noise in the area where it is deployed. It has been operating uninterruptedly for one and a half years since its deployment, except for maintenance periods.

MASE 1.0 was deployed in March 2020 on a scientific buoy which is part of the CanBIO project, promoted by the Government of the Canary Islands and Loro Parque Foundation. The goal of this project was to study the effects of climate change on marine biodiversity in the Canary Islands and Macaronesia. This focusses on three study fields: ocean acidification, marine acoustic pollution, and its effect on marine fauna and the loss of marine biodiversity and ecosystems. These studies require the collection of different types of data. A scientific buoy for data collection, called Morgan, which is shown in Figure 1, was installed in Gando Bay in Gran Canaria (Lattitude: 27.929780 and Longitude: −15.3647185). The buoy is equipped with four solar panels with their corresponding charge manager and battery. This provides power for a total of five sensors for the studies that are conducted in the CanBIO project. A weather station, a ${\mathrm{CO}}_{2}$ sensor, a fluorometer, a pH meter, and the MASE instrument.

The remainder of this article is structured as follows. In Section 2, the design and implementation of the MASE instrument is described and in addition, the software design and development of the indices implemented in the instrument are presented. Section 3 shows the results obtained during one and a half years of data collection. In addition, some conclusions on the changes observed in the soundscape studied are presented. Section 4 presents conclusions about the instrument and some comparisons with other similar products, as well as future directions for the development of this system.

## 2. Materials and Methods

This section describes the internal structure and design of MASE 1.0. First, we will describe the hardware design and the mechanical housing. After that, we describe the ecoacoustic indices that are implemented on the instrument and their method of implementation.

### 2.1. Hardware Design

Figure 2 shows the proposed hardware structure of the MASE instrument.

Electronics focuses on supporting the central processing system, in this case a Raspberry Pi 3B+ [35]. We installed the *Pi OS* [36] operating system on the storage system. This makes it easy to develop and deploy different software components that will be executed to acquire, process, send, or store the processed data gathered by our instrument. It also facilitates interaction with the peripherals connected to the device through different interfaces (I2C, I2S, CSI, and DSI). Data processing is done on the ARM Cortex-A53 1.4GHz processor that is provided with 1GB RAM.

Data acquisition is carried out with a 24-bit acquisition card (HiFiBerry DAC + ADC, [37]) which is configured to run at a sampling rate of up to 192 kHz. The HiFiBerry is connected to the hydrophone (Aquarian AS-1) through the preamplifier (Aquarian PA4-Board [38]). The Aquarian AS-1 hydrophone has a linear range from 1 Hz to 100 kHz ± 2 dB and its sensitivity is −208 dBV re 1 μPa. The hydrophone signal must be conditioned by a preamplifier. Here, we use an Aquarian PA-4 Board [38] that has a jumper selectable gain from 6 dB to 56 dB and is ready to be used with the Aquarian AS-1 hydrophone [39].

As each measurement should be associated with a timestamp and the Raspberry Pi 3B+ does not provide a built-in real-time clock (RTC), a DS1307 real-time clock (RTC) was added. The circuit is provided with a battery that allows keeping the date and time values while the instrument is switched off.

The results could then be sent over the integrated communication interfaces (Wi-Fi, Ethernet port and Bluetooth) provided by the Raspberry Pi 3B+.

The whole system must be powered with a 12-volt voltage source with at least 2A current. In our case, this comes from the central battery installed on the buoy, which is shared with other equipment installed on it. Circuits are protected with a 4 A fuse to prevent short circuit damage. The 12 V from the external power supply are fed into the internal power circuit, which call the LM7805-723 board. This board was custom designed and manufactured to be incorporated into the instrument. It is based on two regulators that work in a dual way: on the one hand, it feeds the uninterruptible power system (UPS), labelled as J4H-5V-USB [40] system connected to the Raspberry PI 3B+, with a voltage of 5.25 V and a current up to 3A thanks to the linear regulator KA78T05 and a parallel transistor [41]. On the other hand, a μA723u regulator is used to supply the hydrophone preamplifier with a voltage of 8V [42]. The UPS allows the safe shutdown of the instrument when external power is disconnected, preventing critical failures of the system due to storage corruption.

### 2.2. Housing and Mechanical Components

Polyvinyl chloride (PVC) elements were used to manufacture the external casing of the instrument. In addition, a system of trays manufactured using 3D printing was developed to hold the electronics inside MASE.

The design of MASE housing was conditioned by the mechanical structure of the scientific buoy. The instrument must be designed to fit into one of the bays for the instrumentation provided on the buoy. We designed the cover to satisfy this requirement using the commercial off-the-shelf parts listed in Table 1. An isometric view of the design is shown in Figure 3.

The main structure of the cover was constructed using PVC components (main tube, upper blind cap, bottom cap and pipe sleeve). The upper closure of the instrument was performed using 8 steel bolts resistant to marine environments, 8 nuts, and 16 washers. A gasket was used to seal the joint between the two PVC elements to prevent water from flowing into the package. Furthermore, to enable the exit of cables from inside the instrument, 20 mm cable glands, placed in the upper cap, were used, as shown in the image in Figure 3.

To insert the electronics described in Section 2.1 into the casing, a tray system was developed. This system was constructed using a Prusa Hephestos i3 3D model printer [43]. The material chosen to print the different elements was polylactic acid (PLA), which has adequate strength and flexibility to support the weight of electronic elements. Figure 4 shows the design of the support tray.

On this tray, electronic elements are placed, such as the computer system, the analogue–digital converter, or the internal power circuit. In addition, to facilitate the connection of external elements, such as the hydrophone, the communications cable, or the power supply, the necessary connectors were placed in the upper part. To extract or insert the electronics into the cover, the tray was equipped with a handle.

Finally, to fix the instrument to the buoy structure, a double-clamp was used. This double clamp consists of two parts joined by four bolts and their corresponding nut and lock washers. In addition, to fix it to the scientific buoy, another four bolts, nuts, washers, and a pressure washer are used.

### 2.3. Ecoacoustic Indices

In [15], the authors described 14 acoustic indices calculated at one-minute intervals. They defined an acoustic index as a statistic that summarises one aspect of the structure and the distribution of the acoustic energy and information in a recording.

In this first version of the MASE instrument, we implemented three acoustic indicators: spectral energy, ACI [13], and $H_{f}$ [15]. Each of these is computed for the octave bands of the spectra. To compute them, the first step is to calculate the fast Fourier transform of the signal using the windows of *N* samples without multiplying by any window function such as Hann, Turkey, or similar. Let $F_{n}\left[k\right]$ be the Fourier coefficient for the *k*th frequency channel of the *n*th window of *N* samples that results when computing the FFT. The next step is to calculate the energy index, $I_{n}\left[k\right]=\left|\right|F_{n}\left[k\right]{\left|\right|}^{2}$, for each of the $\frac{N}{2}$ frequencies. Then, these spectral energy indices are used to compute ecoacoustic indices. They are computed using *M* non-overlapping windows, covering a time interval of M×N samples, and accumulating the frequencies corresponding to each discrete octave band bin, *b*, ranging from 0 to $\log_{2}\left(N/2\right)-1$. These are the expressions for each of the three indices:Spectral energy in octave bands:
(1)$$E\left[b\right]=\left\{\begin{array}{cc} \sum_{n=0}^{M-1}I_{n}\left[0\right] & b=0\\ \sum_{k=2^{b-1}}^{2^{b}-1}\sum_{n=0}^{M-1}I_{n}\left[k\right] & b\in 1,\log_{2}\left(N/2\right)-1 \end{array}\right.$$Acoustic complexity index (ACI) in octave bands:
(2)$$ACI\left[b\right]=\left\{\begin{array}{cc} \frac{\sum_{n=0}^{M-1}\left|I_{n}\left[0\right]-I_{n-1}\left[0\right]\right|}{\sum_{n=0}^{M-1}I_{n}\left[0\right]} & b=0\\ \sum_{k=2^{b-1}}^{2^{b}-1}\frac{\sum_{n=0}^{M-1}\left|I_{n}\left[k\right]-I_{n-1}\left[k\right]\right|}{\sum_{n=0}^{M-1}I_{n}\left[k\right]} & b\in 1,\log_{2}\left(N/2\right)-1 \end{array}\right.$$$H_{t}$ index in octave bands:
(3)$$H_{t}\left[b\right]=\left\{\begin{array}{cc} -\overset{M-1}{\underset{n=0}{\sum }}\left(\frac{I_{n}\left[0\right]}{\sum_{n=0}^{M-1}I_{n}\left[0\right]}\right)\times \log_{2}\left(\frac{I_{n}\left[0\right]}{\sum_{n=0}^{M-1}I_{n}\left[0\right]}\right)\times \log_{2}{\left(M\right)}^{-1} & b=0\\ -\overset{2^{b}-1}{\underset{k=2^{b-1}}{\sum }}\sum_{n=0}^{M-1}\left(\frac{I_{n}\left[k\right]}{\sum_{n=0}^{M-1}I_{n}\left[k\right]}\right)\times \log_{2}\left(\frac{I_{n}\left[k\right]}{\sum_{n=0}^{M-1}I_{n}\left[k\right]}\right)\times \log_{2}{\left(M\right)}^{-1} & b\in 1,\log_{2}\left(N/2\right)-1 \end{array}\right.$$where:
*N* = window size;*M* = number of windows;$F_{n}\left[k\right]$ = Fourier coefficients;$I_{n}\left[k\right]$ = Fourier coefficients;*b* = octave band bin.


The previous computations of the N/2 Fourier indices allows the efficient computation of the three indices by octave bands, since each index is only part of one of the bands, denoted by its bin index, *b*. Furthermore, this enables computation in real time because each time a data buffer of *N* samples arrives from the sound card, a partial result of the sums in the previous equations can be computed. After *M* windows have been processed, the resulting index for each octave band is obtained and the process starts again to compute the indices for a new time interval.

### 2.4. Data Processing Software

The data processing system is the core of MASE because it transforms raw acoustic data into ecoacoustic indices and energy data that characterise the soundscape in the area of operation. The goal of MASE is to produce these ecoacoustic indices in real time. Furthermore, this is also convenient to reduce the amount of data that must be transmitted. In the case of the instrument deployed in this study, the indices are generated at 10 min intervals.

The processing software was developed in the C/C++ and Python programming languages. We use the GStreamer multimedia framework to obtain a modular development [44] running on the Raspbian operating system [36] installed on the Raspberry Pi 3B+.

GStreamer allows to develop different software plugins that contain one or more GStreamer elements. These elements are software pieces that perform some kind of processing task (e.g., filtering, scaling, encoding, …) on the data taken on its input pads and send the results to the next component over its output pads. Elements are connected to each other through the pads which define their input/output interface. Additionally, each element can have properties that can be written to configure the behaviour of that element. The GStreamer framework provides a rich set of plugins that contain several elements that can be used for a multimedia processing task [45]. However, it also allows the development of custom plugins and elements. GStreamer-based applications are then constructed by interconnecting elements, which perform a concrete task to accomplish a more complex processing task. Such an interconnection is called a pipeline.

Figure 5 shows the pipeline developed to calculate the three ecoacoustic indices and energy in the octave bands defined in Section 2.3, send them over the network, and store them in MASE’s internal storage in real time.

Each box in Figure 5 represents a GStreamer element. The elements with a white background in the diagram are the elements that transform the input data into output data, while elements with grey background are elements that do not perform real processing tasks but perform data distribution and queueing (Tee, Multiqueue) or enforce certain input/output capabilities (e.g., data types, sampling rates, …) on the elements they are interconnecting (capsfilter). In this pipeline, we have mixed elements provided with GStreamer (grey background) and custom made elements (white background). The first element is an alsasrc which takes the data from the audio input hardware (HiFiBerry); this element feeds a raw audio stream into deinterleave to separate the two stereo channels. The data from the channel connected to the hydrophone are then fed into audioconvert to transform the 32-bit integer audio samples into a 32-bit floating point. The scale element allows to add an offset and to scale the incoming data with calibration parameters if they are provided. This is then fed into the spec element. This element is a custom element that computes the fast Fourier transform (FFT) on the data windows of a certain size (*N*) that is configurable through the properties of the element. The output are the energy indices $I_{n}\left[k\right]$ mentioned in Section 2.3. Now, the pipeline bifurcates into three parallel paths. Each path computes the internal sums over *n* in Equations (Equation 1)–(Equation 3), respectively. This is performed by SpectrumAverage, ACISpec and Hspec elements. The number of windows to be accumulated is configurable. Each of the three elements then feeds its output into an OctaveBands element, which computes the sum over *k* to obtain the $\log_{2}\left(N/2\right)$ octave bands. The outputs produced by the three parallel paths are then concatenated by the BufferFussion element. After that, they are fed into the Log element, which allows the data to be written into a log file. The output is then fed to DayLogger and to filesink. The former writes the data into the local storage in a directory structure indexed by date, while the latter writes them into a Linux pipe from which it is then sent to the remote storage system over RS232, Ethernet, or any other implemented transmission system.

### 2.5. Electrical Power and Data Communication

The buoy is supplied with a central unit provided by a third-party developer who is responsible for providing power and communication for the set of instruments installed on the buoy. Power is obtained with solar panels that can be seen in Figure 1 and data communication is carried out with a 3G modem integrated within the central unit. Communication between the instruments and the central unit was performed through the RS-232 protocol. Later, the central unit sends the captured data over Internet to a data collection server that allows them to be analysed.

The instrument can be configured to send a set of indices at certain time intervals, which depend on the *M* and *N* parameters explained in Section 2.3. If each of the $3\times \log_{2}\left(N\right)$ indices are stored in a 32-bit floating-point number and such a set is generated every N×M samples, supposing that samples are 32 bit floating-point numbers, the data reduction factor, *R*, obtained by sending the indices instead of the whole raw data stream is given by the expression in Equation (Equation 4).
(4)$$R=\frac{3\times \log_{2}\left(N\right)}{N\times M}$$
Therefore, the calculation of ecoacoustic indices is useful as a data reduction technique to save the communication bandwidth. The communication system of the buoy must guarantee enough transmission speed to extract the data. Using typical values for *N* and *M*, an enormous reduction in the data to be sent takes place, so that it is easy for the provided communication system to guarantee this.

Regarding the electrical power, the power supply of the whole buoy must guarantee that it can provide the power needed by the MASE instrument and other devices attached to the scientific buoy. Section 3.2.1 shows a set of experimental results to determine the power consumption of the instrument, so that the requirements of a power supply for the MASE instrument can be set.

## 3. Results and Discussion

After assembly and construction, the MASE instrument was deployed to the buoy in Gando Bay in March 2020. It has been continuously operating and providing soundscape data for more than a year, except for a period of 4 weeks in November 2020 during which the complete buoy was removed from its location and brought to land to perform maintenance and clearance operations.

### 3.1. Instrument Assembly and Deployment

The final appearance of the instrument, after manufacture and assembly, can be observed in Figure 6. Taking into account the weight of each of the elements in Table 1, the outer casing of the acoustic instrument has a total weight of 12.3 kg. The weight of the tray structure is approximately 500 grams. With the electronic elements attached, the internal tray weighs approximately 1.2 kg. The total weight of the instrument is approximately 14 kg. The total material cost of the instrument is less than EUR 1500.

After assembly, MASE was placed in the buoy’s bay and connected to the power and communication system available on the buoy. The position of the instrument installed on the buoy can be observed in Figure 1. The hydrophone is attached to the anchor chain that connects the buoy to the sea floor. It was placed at a depth of approximately 7 m.

### 3.2. System Operation Parameters

As explained above, the instrument has been operating for more than a year, except for three time gaps that correspond to maintenance and other operations. To achieve this, two aspects must be considered: current consumption and operating temperature.

#### 3.2.1. Current Consumption

As can be observed in Figure 1, the buoy is powered with solar energy. In this situation, it is very important to control the power consumption of the instruments placed on the buoy to prevent more power being consumed than can be produced and stored in the battery system allocated on the buoy. For this reason, during the instrument test phase, power consumption must be properly characterised and measured. Figure 7, shows the consumption of the instrument during the initial setup and in the subsequent steady-state operation phase. A starting peak related to the recharging of the super capacitors of the uninterruptible power supply was introduced in Section 2.1. Fifty seconds later, after the initial peak, the average consumption was approximately 0.70 A, which represents approximately 8.4 W of power. However, some spikes can still be observed due to the switching of the charging circuit of the UPS capacitors, which require a large amount of current.

We consider that the consumption of the instrument is quite acceptable considering the subsystems it contains. With these consumption curves, it has been possible to maintain the continuous operation of the instrument from the moment of its installation to the present without the failure of the batteries mounted on the scientific buoy.

Consumption could be further reduced by directly powering the Raspberry Pi 3B+ at 12 V, avoiding the voltage conversion made by the power system mentioned in Section 2.1. This requires another model for the UPS, but in that case, the estimated power consumption would be approximately 0.4 A with a start-up peak of 1 A, as tested in a second iteration of the instrument’s power supply system.

The knowledge of the actual power consumption of the MASE instrument enables the minimum specifications of the power supply of scientific buoy to be determined. We verified this using a theoretical model of solar irradiance in the area of deployment and the specifications of the solar panels and the battery installed on the buoy, achieving satisfactory results.

#### 3.2.2. Operating Temperature

To ensure the proper operation of the system, it is important that the Raspberry Pi CPU is kept in safe operating temperature ranges. Otherwise, the system may not work properly or simply stop working. The limit indicated by the manufacturer of the Raspberry Pi is 85 ${}^{\circ }$C. To monitor the CPU temperature and to derive insight into the behaviour of the temperature inside of the cover during operation, we decided to record the value of the CPU temperature and to send it joint with acoustic data. Figure 8a shows the evolution of the temperature during the day and during April 2021. The maximum temperature for the CPU is reached approximately between 9:00 and 18:00, which coincides with the hours of most exposure to sunlight. Similar behaviour can be observed in Figure 8b, which shows the outside temperature at the location of the buoy. As expected, the temperature inside of the instrument is related to the outside temperature. Fortunately, the casing of the acoustic instrument is in direct contact with seawater, so it is possible to dissipate some of the heat generated within the acoustic instrument. The temperature reached by MASE during its operation time at sea was below the limit, which means that the package design and dissipation are suitable to keep the instrument in operation.

### 3.3. Instrument Measurements

Figure 9 summarises the soundscape over a time period of more than one and half years. It is a false colour spectrogram [15] which shows the activity in the different logarithmic frequency bands. The red channel shows the spectral energy, the blue channel shows the ACI index, and the blue channel corresponds to the $H_{t}$ index, as defined in Section 2.3. The black areas correspond to non-operation periods due to maintenance or other contingencies. The indices were retrieved from the MASE instrument during its operation time. For the purpose of representation, we performed a histogram equalisation of each index image and normalised it between 0 and 1. The instrument was configured to throw a set of indices every 10 min, which corresponds to a vertical slice in the false colour spectrogram. This was achieved by setting the configuration parameters *N* and *M* described in Section 2.3. The value of *N* was set to 4096 while the value of *M* was set to 28,125. The sampling rate of the sound card was fixed to 192 kHz.

It can be observed that the soundscape description provided by these three indices has changed over time, allowing to distinguish three main periods of time. Such a change of activity can be observed between the periods from April to June 2020 and the values measured from August to November 2021 at low frequencies. The former period was during the lockdown in the Canary Islands due to the pandemic and the latter was the beginning of the end of COVID-19 restrictions and the first anthropogenic activity. Thus, it can be seen that during a period when anthropogenic noise introduced by ships in the area is not present, the soundscape and consequently the instrument’s measurements are different. Other changes are also observable in Figure 9. To relate them with a concrete cause, long time observation experiments need to be performed in the area.

A detailed comparison between the indices obtained during the pandemic lockdown and those caught during the re-establishment of maritime activity can be seen in Figure 10. The left column shows the evolution of the three indices during a period of time within the lockdown, while the right column shows the evolution of the indices during a period of time after the lockdown. As can be observed, the spectral energy as well as the ACI index show significant variation in sound levels. For the $H_{t}$ index, the values remain somewhat more similar, so that in the case of the study, the two previous indices seem to provide more information. Noticeable variations in the octave bands between 23 and 4.5 kHz can be observed. Within this range of bands are those included in indicator 11.2.1 of the European Marine Strategies Directive. This appreciable change may be due to anthropogenic activity in the area of study. Furthermore, periodic behaviour can also be observed in the right column, where more anthropogenic activity is expected, which coincides with day and night cycles.

Integrating soundscape information into a single graph such as that in Figure 9 reduces the amount of data needed for the analysis of that soundscape by detecting changes. In general, to obtain such a graph requires analysing large raw acoustic data files in an off-line processing and with a latency of some months. However, MASE is able to process the audio in real time and send them to a remote data server every 10 min. As such, MASE enables the analysis of soundscapes in an effective and efficient way, which is able to obtain these processed data from the remote scientific buoy while they are produced. Furthermore, as the acoustic instrument is reprogrammable, it is possible to implement as many indices as necessary to characterise the soundscape to be studied. This would allow extending the set of acoustic indices to obtain deeper insight into the soundscape of the area of study.

## 4. Conclusions and Future Work

MASE is an instrument that allows underwater soundscape analysis by synthesising ecoacoustic indices from the raw audio signal. The computation of the ecoacoustic indices is performed in real time which allows monitoring the soundscape in the area and to react to unexpected events or drastic changes. The ecoacoustic indices are sent over the Internet with low bandwidth consumption which allows performing an almost live monitoring of changes in the soundscape that can be registered in the range of the sensor (acoustic range of the hydrophone). This could be performed by other systems, but in a delayed manner.

The implementation of the acoustic index calculation in real time using the GStreamer framework results in a very low consumption of CPU and memory resources. There is still room for more complex processing methods. This makes MASE a versatile tool for running more complex algorithms such as acoustic event detection or inference using a neural network.

As shown in Section 3, MASE was able to work uninterruptedly, as well as maintain adequate temperature inside while preserving its structural integrity during long periods of time. This allowed the monitoring and the characterisation of the soundscape of the test area in Gando Bay without the need for human interventions to access the data. Further data integration on the server side allows long-term soundscape monitoring, where changes in the soundscape can be observed (Figure 9 and Figure 10). The instrument has allowed us to compare the period of human-induced acoustic inactivity due to lockdown during the pandemic and the return of anthropogenic noise activity.

Although it shows similarities with [34], to our knowledge, MASE is one of the first instruments that provides information about underwater soundscape in real time using ecoacoustic indices, so it does not need a server or external equipment to perform data processing. On the other hand, it requires a less powerful power supply system and can be adapted to different types of mounting as it is a modular design. Finally, in comparison to the instrument presented in [34], MASE has an integrated analogue–digital interface within the system which allows lower power consumption, a higher number of quantisation bits, and a higher sampling rate. This allows the instrument to also be used for the acoustic monitoring of cetaceans and other species that perform vocal activity in the ultrasonic band. In this way, MASE can be adapted to different ecoacoustic study scenarios.

In the future, due to the modular hardware and software design described in Section 2, it is possible to improve the individual components of the instrument by adding more functionalities or extending existing ones. The set of ecoacoustic indices provided by MASE will be extended so that more exhaustive soundscape information can made be available to scientists and administrations. The low-cost off-the-shelf components upon which MASE is built make it more suitable for deploying several of these instruments to characterise the soundscape not only of a specific area but also of a coastline, the entrance of a port, etc. The possibility of adapting the software and external packaging makes it a versatile tool for the study of underwater soundscapes.

The current version of MASE is not provided with its own communications system, because it would be redundant with that provided on the scientific buoy itself and which integrates the measurements of all the sensors and sends them to the remote storage. An updated version of MASE equipped with its own communications system, based on LoRa, is currently being tested. This allows one to increase the amount of data sent by the instrument and the access as an Internet of Things device.

## Figures and Tables

**Figure 1 sensors-22-03404-f001:**
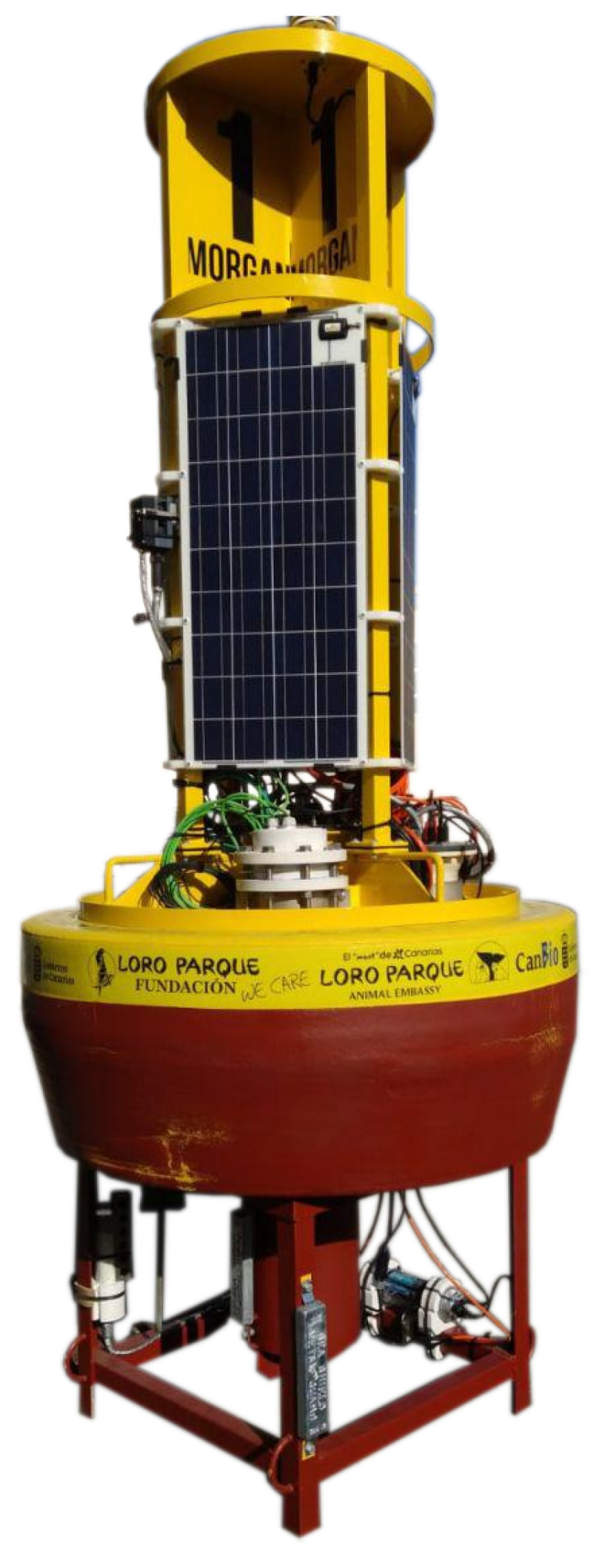
Morgan buoy.

**Figure 2 sensors-22-03404-f002:**
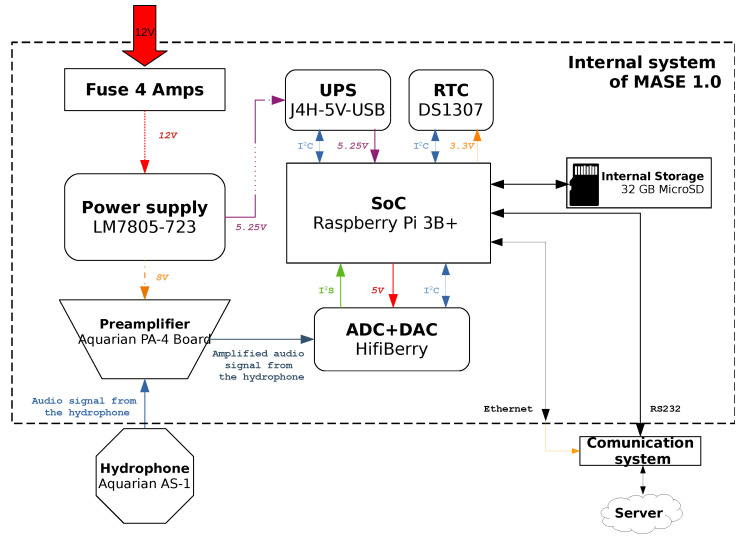
Proposed hardware structure for MASE.

**Figure 3 sensors-22-03404-f003:**
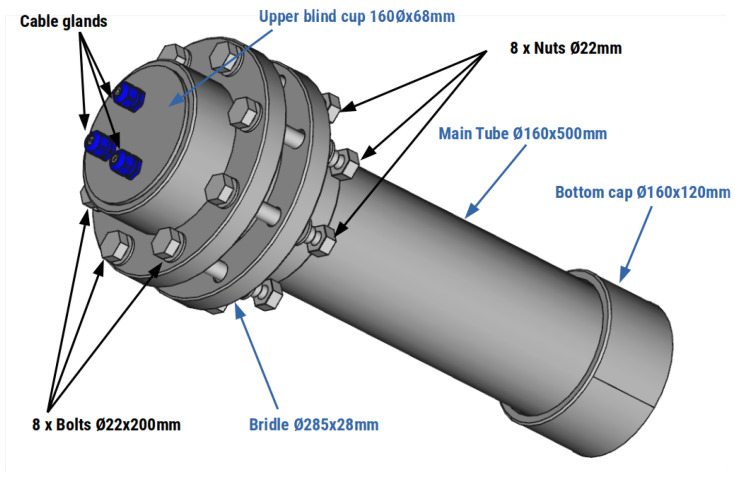
Isometric view of the MASE’s housing.

**Figure 4 sensors-22-03404-f004:**
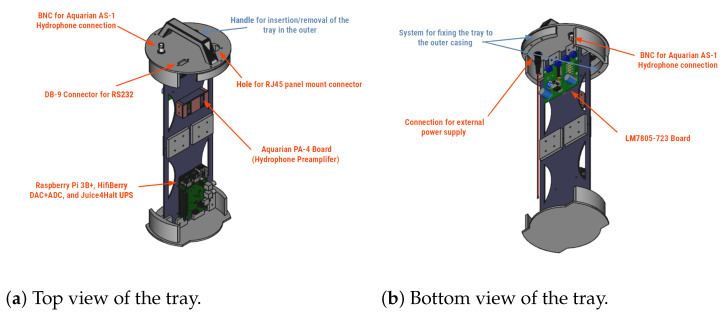
MASE internal system tray.

**Figure 5 sensors-22-03404-f005:**
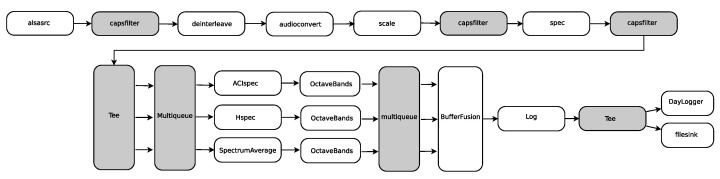
GStreamer pipeline for the computation of ecoacoustic indices and energy in octave bands.

**Figure 6 sensors-22-03404-f006:**
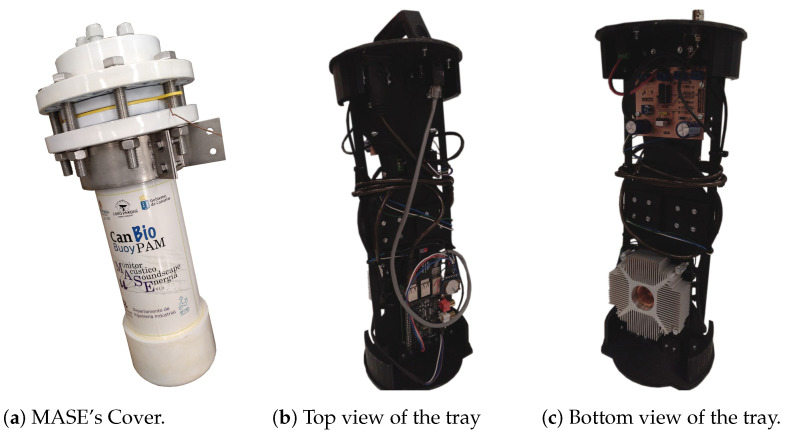
MASE’s final appearance.

**Figure 7 sensors-22-03404-f007:**
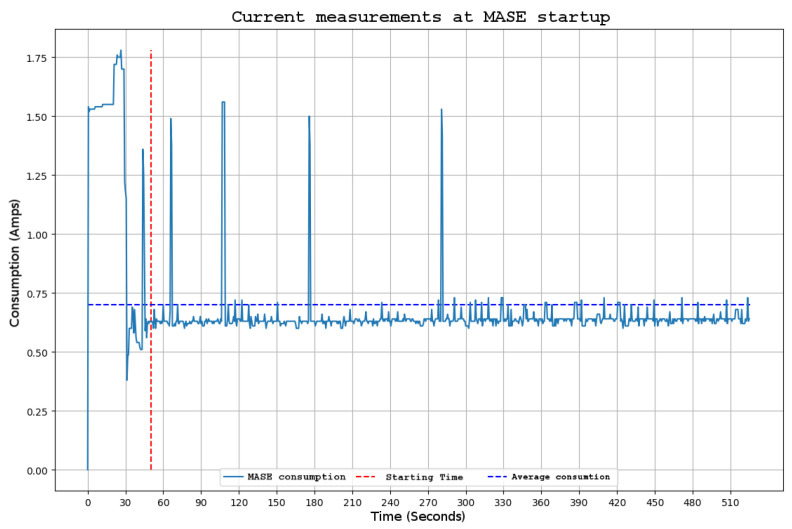
MASE consumption during start-up and running.

**Figure 8 sensors-22-03404-f008:**
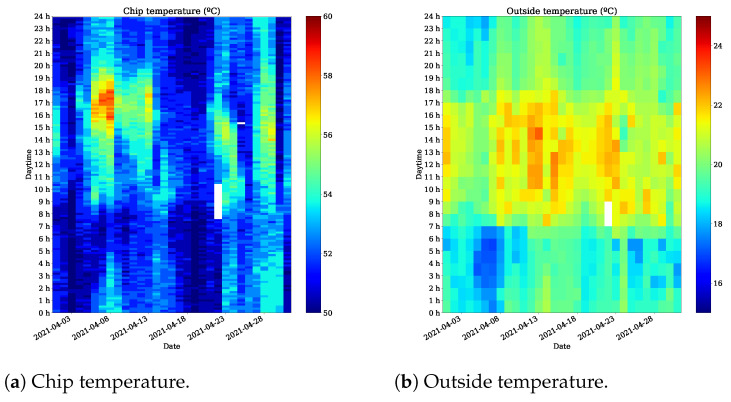
Temperature evolution (April 2021).

**Figure 9 sensors-22-03404-f009:**
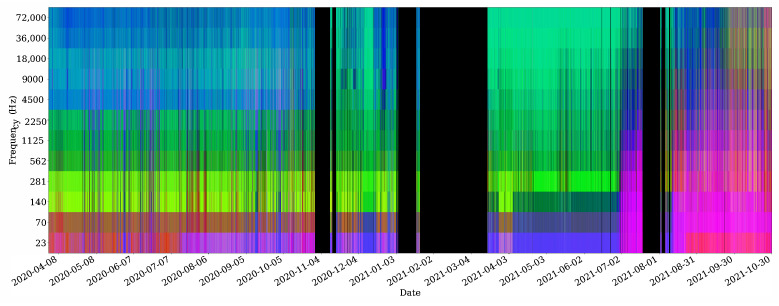
Acoustic indices implemented in MASE (ACI, $H_{t}$, and energy spectrum) integrated in a single graphic as false colour spectrogram.

**Figure 10 sensors-22-03404-f010:**
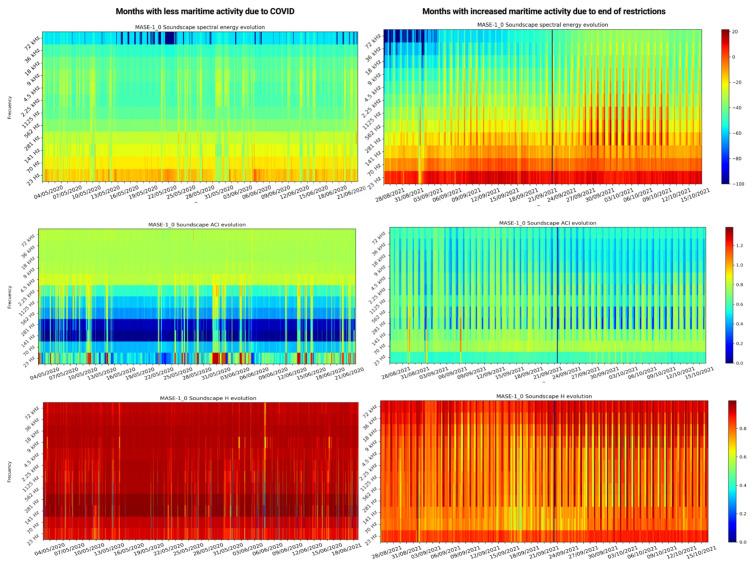
Indices obtained during confinement vs. reactivation of maritime activity (energy spectrum, ACI, and $H_{t}$).

**Table 1 sensors-22-03404-t001:** Materials for the construction of the cover (where ⊘ is the diameter).

Part	Quantity	Material	Size (mm)
Main tube	1	PVC-U	⊘160 × 500
Upper blind cap	1	PVC-U	⊘160 × 68
Pipe sleeve	2	PVC-U	⊘214 × 17 + ⊘190 × 79
Bridle	2	PVC-U	⊘285 × 28
Flat gasket	1	PVC-U	⊘160 × 4
Bottom cap	1	PVC-U	⊘160 × 120
Bolt	8	A4 Stainless Steel	⊘22 × 200
Nuts	8	A4 Stainless Steel	⊘22
Washers	16	A4 Stainless Steel	⊘22

## Data Availability

Not applicable.

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
