# Peer review of "MASE: An Instrument Designed to Record Underwater Soundscape"

_sensors, 2022, doi:10.3390/s22093404_

Round 1

Reviewer 1 Report

Please to read the attached file, thank you.

Reviewer 3 Report

In this paper, the authors propose a low-cost instrument to monitor soundscape in a marine area using ecoacoustic indices or other strategies. In general, the topic is very important for acoustic signal processing. However, this paper just presents the authors’ product. The authors should highlight their work before acceptance. Specifically, the advantages of hardware or software compared to traditional ones should be focused.

Based on the authors’ paper, this paper aims to introduce their product. However, the authors cannot comprehensively describe their product. From reviewer’s view, processing results, advantages compared to traditional product, etc, should be introduced.

In this paper, the authors cannot highlight their advantages of product. Nowadays, the non-Gaussian underwater noise in shallow water is studied by many researchers. Up till now, many papers such as following papers about non-Gaussian underwater noise have been published. However, the collection of non-Gaussian underwater noise requires that the equipment has high performance. The authors should clarify this issue in their paper.

X Zhang,et al.Parameter estimation of underwater impulsive noise with the Class B model.IET Radar, Sonar & Navigation,2020,Doi: 10.1049/iet-rsn.2019.0477.

Mahmood, et al, Modeling Colored Impulsive Noise by Markov Chains and Alpha-Stable Processes, in OCEANS 2015 MTS/IEEE, (Genoa, Italy), May 2015.

In section 3, the authors should present their processing results based on their product. Then, the readers can visually follow the authors’ work and product.

Round 2

Reviewer 3 Report

The paper is improved after revision, and the reviewer has no more comments.